# Fertilizer $^{15}$N Fates of the Coastal Saline Soil-Wheat Systems with Different Salinization Degrees in the Yellow River Delta

**Kongming Zhu** [1] , **Fupeng Song** [1,*] , **Fujian Duan** [1], **Yuping Zhuge** [1,*] , **Weifeng Chen** [1], **Quangang Yang** [1], **Xinsong Guo** [2], **Pizheng Hong** [2], **Li Wan** [3] and **Qun Lin** [1]

1   National Engineering Research Center for Efficient Utilization of Soil and Fertilizer, College of Resources and Environment, Shandong Agricultural University, Tai'an 271018, China
2   Engineering Technology Research Center of Shandong Province for Efficient Utilization of Humic Acid, Shandong Nongda Fertilizer Science and Technology Co., Ltd., Feicheng 271600, China
3   Shandong Ludong Road and Bridge Co., Ltd., Dongying 257000, China
*   Correspondence: fpsong@126.com (F.S.); zhugeyp@sdau.edu.cn (Y.Z.)

**Abstract:** In order to clarify the fates of fertilizer N in coastal saline soil-wheat systems with different salinization degrees, this study was conducted to determine the $^{15}$N uptake rates in various parts of wheat plant at maturity stage and the residual $^{15}$N in three different saline soils and the $^{15}$N loss of soil-wheat systems by using the $^{15}$N-labeled urea N tracing method in the Yellow River Delta. The results showed that: (1) The increase of soil salinity from 0.2% to 1% promoted the wheat plant to absorb N from soil and not from fertilizer and significantly inhibited the dry matter mass accumulation and $^{15}$N uptakes of each wheat parts and whole plant, but especially increased the total N concentration of wheat roots, stems, leaves, and grains. The aggravation of soil salinity significantly enhanced the distribution ratios of $^{15}$N uptakes and Ndffs in the wheat roots, stems, and leaves to depress the salt stress. (2) The $^{15}$N residues were mainly concentrated in the 0~20 cm saline soil layer and decreased as the soil profile deepened from 0 to 100 cm; the $^{15}$N residues decreased in the 0~40 cm soil profile layer and accumulated in the 40~100 cm with the increase of soil salinization degrees significantly. (3) The fates of $^{15}$N applied to the coastal saline soil-wheat system were wheat uptakes 1.53~13.96%, soil residues 10.05~48.69%, losses 37.35~88.42%, with the lowest $^{15}$N uptake and utilization in the three saline soils, the highest residual rate in lightly saline soils, and the highest loss in moderately and heavily saline soils. The increase of soil salinity inhibits wheat uptakes and soil residues and intensifies the losses from fertilizer $^{15}$N. Therefore, the fate of fertilizer N losses significantly increased as the degree of soil salinity increased. The conventional N management that was extremely inefficient for more N loss should be optimized to enhance the N efficiency and wheat yield of the coastal saline soil-wheat system in the Yellow River Delta.

**Keywords:** fertilizer $^{15}$N fate; coastal saline soil-wheat system; N uptake; N residue; N loss

## 1. Introduction

Soil salinization is a widespread problem on more than 25% of the global soil area, and more than 36 million hectares of land in China are affected by salinization, representing about 4.9% of all available land [1–7]. The Yellow River Delta, located in the warm temperate subhumid continental monsoon climate zone, has experienced serious soil salinization from the coastline to the interior over the past 20 years due to seawater intrusion [8–12]. The coastal saline soils in the Yellow River Delta are an important back-up land resource for food production in China [13]. However, shallower saline groundwater, high soil pH and salinity, poor soil nutrients and structure, low water retention capacity, and other problems of the coastal saline soils reduce crop growth and yields [14–17], ultimately constraining sustainable development of agriculture and food security of the Yellow River Delta.

Nitrogen is a basic nutrient required for plant growth and development, and it is one of the especially deficient nutrients in the coastal saline soil of the Yellow River Delta [18].

Nitrogen fertilizers have been widely applied to increase crop yields in the coastal saline soil to address the pressure on the increasing food security due to the population boom in this century [19]. Urea is commonly used as a source of nitrogen fertilizer for its high nitrogen content and stability [20,21]. The conventional amount of urea application in Chinese crop production is 2–3 times higher than that of the world averagely [22]. However, less than 30% of urea nitrogen is absorbed by crops in China, while more than 70% leaches into the soil profile and groundwater with rainfall and irrigation, or runoffs into surface water resulting in eutrophication of water bodies [23,24].

Wheat is one of the major food crops grown in the costal saline soil of the Yellow River Delta and its growth is affected by both N deficiency and salt stress [25]. Currently, many studies have focused on increasing the application amount of the nitrogen fertilizer to close yield gaps in the saline soil, but the effect may be limited by the soil saline environment. Meanwhile, nitrogen deficiency of coastal saline soil increases the severity of salt stress [26–28]. So, nitrogen fertilizer management is an important and urgent task of agricultural production in coastal saline soil.

Stable $^{15}$N isotope labeling is a well-established technique for tracking and quantifying N transformations mostly applied in various ecosystems and in non-saline soil environment [29–32]. This technique is also used in some studies of saline soil, but mainly applied to specific saline soil [33], and less used in different saline soils, especially in soil-wheat systems with different degrees of salinization, which has less been reported.

Compared to most of previous studies, which are still confined to indoor pot conditions, this technique with $^{15}$N-labeled urea in field is more reflective of the actual fate of urea N in coastal saline soil-wheat systems with different salinization degrees [34]. Therefore, the objectives of this research are to quantify the N uptake by wheat plants at maturity, the N residual distribution and the N loss in the coastal saline soil-wheat system with different degrees of salinity by $^{15}$N-labeled urea tracing method, and to provide a theoretical basis for improving nitrogen management in the wheat planting process which may contribute to the sustainable agriculture development of the Yellow River Delta.

## 2. Materials and Methods

### 2.1. Experimental Site and Soil Properties

The experimental site is located in Bohai farm, Dongying City, Shandong Province (37°47′ N, 118°36′ E), which is in the hinterland of the Yellow River Delta. The topography of the area is flat, and the degree of soil salinization is unevenly distributed. The soil types are mainly coastal chloride saline fluvo-aquic soils with loamy and clay loamy textures. The burial depth of groundwater is 0.8~1.2 m with salinity 12.69~20.08 g L$^{-1}$, which is highly mineralized and not suitable for irrigation. The region has a warm temperate semi-arid continental monsoon climate with an average annual temperature of 13.3 °C, an average annual frost-free period of 206 days, an average annual precipitation of 537 mm and an average annual evaporation of 1885 mm. There is light-saline and medium-salinity land, heavy-saline land, and salty wasteland in this area, accounting for 66%, 18%, and 16% respectively, wheat-corn rotation and cotton are cultivated in saline land except for salty wasteland. The natural vegetation is dominated by halophytes, such as suaeda salsa (*Suaeda salsa* L.), reeds (*Phragmites australis* L.), and tamarisk (*Tamarix chinensis* L.).

The three different salinization experimental coastal saline soils were the light (LS), medium (MS), and heavy (HS) salinity with the classification of the soil water-soluble salt contents: 0.2%–0.4%, 0.4%–0.6%, and 0.6%–1.0%, respectively (China Soil Census Office, 1992), and the basic physicochemical properties are shown in Table 1.

**Table 1.** Properties of light-salinity soil (LS), medium-salinity soil (MS), and heavy-salinity soil (HS).

| Soil | pH | EC (dS m$^{-1}$) | TS (g kg$^{-1}$) | BD (g cm$^{-3}$) | OM (g kg$^{-1}$) | TN (g kg$^{-1}$) | AP (mg kg$^{-1}$) | AK (mg kg$^{-1}$) |
|---|---|---|---|---|---|---|---|---|
| LS | 7.90 ± 0.01 | 0.42 ± 0.02 | 2.97 ± 0.28 | 1.38 ± 0.05 | 15.23 ± 0.19 | 1.05 ± 0.01 | 18.75 ± 0.49 | 185.70 ± 0.80 |
| MS | 7.98 ± 0.01 | 1.24 ± 0.01 | 5.02 ± 0.24 | 1.32 ± 0.05 | 12.72 ± 0.09 | 0.93 ± 0.03 | 12.08 ± 0.10 | 227.05 ± 3.15 |
| HS | 7.99 ± 0.02 | 2.97 ± 0.03 | 6.91 ± 0.55 | 1.30 ± 0.09 | 7.99 ± 0.07 | 0.76 ± 0.02 | 9.33 ± 0.10 | 250.30 ± 8.70 |

Note: EC, soil electrical conductivity; TS, soil total salt; BD, soil bulk density; OM, soil organic matter; TN, soil total nitrogen; AP, soil available phosphorus; and AK, soil available potassium.

### 2.2. Experimental Design

The field experiment of wheat (*Triticum aestivum* L., Jimai 23) was conducted from October 2017 to June 2018 with a field micro-zonal randomized block trial design. The treatments consisted of three fields with different salinization degrees: LS (light-salinity soil), MS (medium-salinity soil), and HS (heavy-salinity soil) and each treatment was repeated three times. The field micro-zones with 1 m$^2$ square area were delimited in the middle area of each field and were separated by 1.5 m distance. The plastic cloth was buried into a ditch dug 0.2 m wide and 0.6 m deep to form a ridge 0.2 m wide and 0.05 m high above the soil surface and surrounding every micro-zone, which prevented the $^{15}$N-labeled urea run offing from each micro-zone.

The conventionally chemical fertilizers were applied with 225 kg N hm$^{-2}$ (urea, N $\geq$ 46.4%)-150 kg P$_2$O$_5$ hm$^{-2}$ (superphosphate, P$_2$O$_5$ $\geq$ 18%)-75 kg K$_2$O hm$^{-2}$ (potassium sulfate, K$_2$O $\geq$ 50%) in the experimental fields. All the superphosphate, potassium sulfate and 1/2 amount of urea were basely applied and mixed with topsoil before wheat sown, another 1/2 amount of urea was top-dressed at the jointing stage of wheat. The amount and method of the chemical fertilizers applied in each micro-zone were as the same as the outside fields except for the $^{15}$N labeled urea (N $\geq$ 46.4%, 2.0% $^{15}$N abundance, produced by Shanghai Research Institute of Chemical Industry, Shanghai, China) which was applied plot by plot before the application of outside field. All the other management measures were consistent with the local farmer practices.

### 2.3. Sampling, Measurement Methods and Assessment of Nitrogen Accumulation

Plant and soil samples were collected during wheat harvest stage in June 2018. All wheat plant samples in the micro-zone were harvested, and the roots, stems, leaves, sheaths, and grains were separated and dried at 60 °C to determine the dry matter masses. Soil samples were collected in 5 layers: 0~20 cm, 20~40 cm, 40~60 cm, 60~80 cm, 80~100 cm.

Soil total salt content was measured with the drying residue method (water-soil ratio 5:1), and soil total N content was determined with the semi-micro Kjeldahl method. Plant total N concentration was determined by H$_2$SO$_4$-H$_2$O$_2$ digestion and the Kjeldahl method. Soil and plant $^{15}$N abundance was determined using a mass spectrometer (Isoprime, Manchester, UK).

The percentage of plant nitrogen derived from fertilizer nitrogen (Ndff, %) was calculated as

$$Ndff = (N_P - N_A) / (N_f - N_A) \times 100 \qquad (1)$$

where N$_P$ represents the $^{15}$N abundance (%) in plant samples, N$_A$ represents the natural abundance of $^{15}$N (N$_A$ = 0.365%), and N$_f$ represents the $^{15}$N abundance (%) in fertilizer.

The percentage of plant nitrogen derived from soil nitrogen (Ndfs, %) was calculated as

$$Ndfs = 1 - Ndff \qquad (2)$$

The plant $^{15}$N uptake (F$_N$, g m$^{-2}$) was calculated as

$$F_N = Ndff \times D_M \times N_C \qquad (3)$$

where D$_M$ is plant dry matter mass (g m$^{-2}$) and N$_C$ is plant total nitrogen concentration (%).

The $^{15}N$ fertilizer utilization efficiency ($^{15}NUE$, %) in the current season was calculated as the wheat $^{15}N$ uptake, dividing the amount of N applied, which is the $^{15}N$ uptake rate:

$$^{15}NUE = F_N / {}^{15}Nap \times 100 \tag{4}$$

where $^{15}Nap$ is the amount of $^{15}N$ applied per square of wheat (g m$^{-2}$).

Soil $^{15}N$ residue ($R_N$, g m$^{-2}$) was calculated as

$$R_N = D_{MS} \times N_{CS} \times ({}^{15}N_S - N_A) \times 100 \tag{5}$$

where $D_{MS}$ is dry weight of soil sample (g), $N_{CS}$ is total soil nitrogen content (%), and $^{15}N_S$ is soil $^{15}N$ abundance (%).

Soil $^{15}N$ residual rate ($R_{Nr}$, %) was calculated as

$$R_{Nr} = R_N / {}^{15}N_{AP} \times 100 \tag{6}$$

$^{15}N$ loss ($L_N$, g m$^{-2}$) was calculated as

$$L_N = {}^{15}N_{AP} - F_N - R_N \tag{7}$$

$^{15}N$ loss rate ($L_{Nr}$, %) was calculated as

$$L_{Nr} = 100\% - NUE - R_{Nr} \tag{8}$$

### 2.4. Statistical Analysis

The data of wheat plant and soil properties from the experimental field were analyzed by one-way analysis of variance (ANOVA) using SPSS 19.0 software (IBM Corporation, Armonk, NY, USA), and multiple comparisons of means were performed using Duncan's test ($p < 0.05$). Data calculation and graphing were respectively implemented with Excel 2016 software (Microsoft Corporation, Redmond, WA, USA) and Origin 2019 software (Origin Lab Corporation, Northampton, MA, USA).

## 3. Results

### 3.1. Dry Matter Mass of Each Wheat Parts

With the aggravation of soil salinization, the dry matter masses of the same wheat parts and whole plant of different salinization soil treatments showed a downward trend (Table 2). There were significant differences of the dry matter masses of the wheat root and stem between MS and LS treatments, HS and LS treatments with no significant differences of these between MS and HS treatments. However, there were significant differences of the dry matter masses of the wheat leaf, sheath, grain, and whole plant among the three treatments.

**Table 2.** Dry matter mass (g m$^{-2}$) of each wheat parts in different treatments.

| Treatment | Root | Stem | Leaf | Sheath | Grain | Whole Plant |
|---|---|---|---|---|---|---|
| LS | 22.63 ± 0.93 a | 125.60 ± 7.64 a | 94.69 ± 3.83 a | 103.26 ± 2.19 a | 222.83 ± 25.61 a | 569.02 ± 18.97 a |
| MS | 3.40 ± 0.43 b | 18.85 ± 1.41 b | 15.37 ± 0.26 b | 29.13 ± 3.34 b | 82.64 ± 6.40 b | 149.38 ± 4.44 b |
| HS | 1.75 ± 0.02 b | 6.95 ± 0.24 b | 6.33 ± 0.13 c | 7.69 ± 0.15 c | 14.80 ± 0.59 c | 37.51 ± 0.52 c |

Note: The values in the table are the mean ± standard error and different letters in each column indicate significant difference ($p < 0.05$).

Compared with LS treatment, HS treatment and MS treatment significantly reduced the dry matter masses of the wheat root, stem, leaf, sheath, grain, and whole plant by 92.28%, 94.47%, 93.32%, 92.56%, 93.36%, and 93.41%; and by 84.97%, 84.99%, 83.77%, 71.79%, 62.92%, and 73.75%, respectively ($p < 0.05$). The dry matter masses of the wheat leaves, sheaths, grains, and whole plants in HS treatment significantly decreased than those

of corresponding parts of wheat plants in MS treatment by 58.83%, 73.61%, 82.09%, and 74.89%, respectively ($p < 0.05$).

The order from high to low of each wheat part dry matter mass in the same saline soil treatment with the same salinization degree were basically in accordance with the order of grain, sheath, stem, leaf, and root, and the dry matter mass of the wheat grain was higher than those of the other parts with 77.41~2328.79%. So, the dry matter masses of wheat plants in saline soils were mainly concentrated in wheat grains than the other wheat parts, and the increase of soil salinization would inhibit the accumulation of the dry matter masses in various wheat parts and the whole plants.

### 3.2. Total N Concentration, $^{15}$N Uptake, Distribution Ratio of $^{15}$N Uptake, Ndff and Ndfs in Wheat Parts

With the increase of soil salinization, the total N concentrations in all wheat parts and the whole wheat plant showed an upward trend (Table 3).

**Table 3.** Total N concentration, $^{15}$N uptake, distribution ratio of $^{15}$N uptake, Ndff, and Ndfs in wheat parts of different treatments.

| Item | Treatment | Root | Stem | Leaf | Sheath | Grain | Whole Plant |
|---|---|---|---|---|---|---|---|
| Total N Concentration/% | LS | 0.74 ± 0.02 c | 0.47 ± 0.02 c | 0.80 ± 0.02 c | 0.84 ± 0.01 b | 1.87 ± 0.01 b | 1.15 ± 0.04 b |
| | MS | 1.02 ± 0.02 b | 0.60 ± 0.02 b | 1.27 ± 0.01 b | 1.33 ± 0.05 a | 2.34 ± 0.11 a | 1.78 ± 0.03 a |
| | HS | 1.13 ± 0.01 a | 0.93 ± 0.03 a | 1.32 ± 0.00 a | 1.47 ± 0.10 a | 2.49 ± 0.19 a | 1.74 ± 0.07 a |
| $^{15}$N Uptake/(g m$^{-2}$) | LS | 0.05 ± 0.01 a | 0.15 ± 0.01 a | 0.32 ± 0.01 a | 0.36 ± 0.01 a | 2.28 ± 0.30 a | 3.15 ± 0.28 a |
| | MS | 0.01 ± 0.00 b | 0.04 ± 0.00 b | 0.08 ± 0.00 b | 0.17 ± 0.01 b | 1.09 ± 0.13 b | 1.39 ± 0.12 b |
| | HS | 0.01 ± 0.00 b | 0.02 ± 0.00 b | 0.04 ± 0.00 c | 0.05 ± 0.00 c | 0.23 ± 0.01 c | 0.35 ± 0.01 c |
| Distribution Ratio of $^{15}$N Uptake/% | LS | 1.59 ± 0.18ab | 4.88 ± 0.72ab | 10.23 ± 1.26 a | 11.66 ± 1.13 a | 71.64 ± 3.17ab | / |
| | MS | 0.70 ± 0.09 b | 2.73 ± 0.45 b | 5.94 ± 0.57 b | 12.59 ± 2.04 a | 78.04 ± 3.06 a | / |
| | HS | 2.16 ± 0.45 a | 7.12 ± 1.11 a | 10.84 ± 1.33 a | 13.97 ± 1.77 a | 65.91 ± 1.66 b | / |
| Ndff/% | LS | 25.52 ± 0.55 b | 28.16 ± 1.00 c | 36.85 ± 0.44 b | 41.56 ± 1.30 a | 47.05 ± 2.21 b | 45.63 ± 1.72 a |
| | MS | 29.53 ± 1.53ab | 32.89 ± 0.86 b | 40.28 ± 0.97 b | 43.50 ± 2.02 a | 55.47 ± 1.39 a | 45.42 ± 1.54 a |
| | HS | 37.44 ± 3.93 a | 37.79 ± 0.56 a | 44.76 ± 1.46 a | 46.13 ± 1.09 a | 57.62 ± 3.15 a | 50.42 ± 2.48 a |
| Ndfs/% | LS | 74.48 ± 0.55 a | 71.84 ± 1.00 a | 63.15 ± 0.44 a | 58.44 ± 1.30 a | 52.95 ± 2.21 a | 54.37 ± 1.72 a |
| | MS | 70.47 ± 1.53ab | 67.11 ± 0.86 b | 59.72 ± 0.97 a | 56.50 ± 2.02 a | 44.53 ± 1.39 b | 54.58 ± 1.54 a |
| | HS | 62.56 ± 3.93 b | 62.21 ± 0.56 c | 55.24 ± 1.46 b | 53.87 ± 1.09 a | 42.38 ± 3.15 b | 49.58 ± 2.48 a |

Note: The values in the table are the mean ± standard error and different letters in each column indicate significant difference ($p < 0.05$).

The total N concentration of the wheat roots, stems and leaves were significantly different among the three treatments ($p < 0.05$). Compared with those of LS treatment, the total N concentrations of the wheat roots, stems, leaves, sheaths, grains, and the whole wheat plant in HS and MS treatments significantly increased by 33.22~98.73% and 25.47~59.12%, respectively ($p < 0.05$). The total N concentrations of the wheat roots, stems, and leaves in HS treatment were higher than those of the corresponding wheat parts in MS treatment with 11.55%, 56.51% and 4.10% ($p < 0.05$). In the three saline soil treatments, the total N concentrations order of each wheat parts in the same soil treatment were obtained according with grain, sheath, leaf, root and stem. Therefore, compared with the other wheat parts, the total N concentrations in wheat grains were significantly highest. The increase of soil salinization increased the total N concentrations in all wheat parts, especially in the wheat root, stem, and leaf in the highest salinization soil to resist salt stress.

The $^{15}$N uptake per unit area of the same wheat parts or the amounts of fertilizer nitrogen absorbed by the same wheat parts were different in different salinization soil treatments (Table 3). Compared with the $^{15}$N uptakes per unit area of each wheat parts in LS treatment, the $^{15}$N uptakes per unit area of the wheat roots, stems, leaves, sheaths, grains, and the whole plants in MS and HS treatment significantly decreased by 80.32% and 84.94%, 75.30% and 83.59%, 74.19% and 88.16%, 52.80% and 86.63%, 51.93% and 89.98%, and 55.82% and 89.03% ($p < 0.05$). In the three saline soil treatments, the $^{15}$N uptake order of each wheat parts in the same soil treatment were all obtained according with grain,

sheath, leaf, stem, and root. The $^{15}$N uptake of wheat grains was higher than that of other wheat parts by 3.72~111.00 times ($p < 0.05$). So, compared with the other wheat parts, the $^{15}$N uptake of wheat grains were all significantly highest in the three saline soil treatments. With the aggravation of soil salinization, the amount of fertilizer nitrogen absorbed by each wheat part and the whole plant decreased significantly.

The distribution ratios of $^{15}$N uptakes (the ratio of $^{15}$N uptakes in each part of wheat to $^{15}$N uptakes in whole wheat plant) in various wheat parts were different in different salinization soil treatments (Table 3). Although the distribution ratio of $^{15}$N uptakes of the wheat leaf in LS treatment was significantly higher than that in MS treatment by 72.22%, those of the wheat roots, stems, sheaths, and grains in LS treatment were not significantly different between MS treatment and HS treatment ($p < 0.05$). The distribution ratios of $^{15}$N uptakes in the wheat root, stem and leaf in HS treatment were significantly higher than those of the corresponding wheat part in MS treatment by 207.91%, 160.68% and 82.37% except for that of the wheat sheath between HS and MS. Meanwhile, that of wheat grain in HS treatment was significantly lower than that in MS treatment by 15.54% ($p < 0.05$). The order from high to low of the $^{15}$N uptakes distribution ratios of various wheat parts in the same saline soil treatment was accordance with the grain, sheath, leaf, stem, and root. Hence, when the degree of soil salinization increased to HS level, the $^{15}$N uptakes distribution ratio of the wheat root, stem, and leaf increased significantly. Meanwhile, there was no effect on the wheat sheath, but that in the wheat grain decreased significantly. The changes of soil salinization degree had no significant effects on the high and low order of the $^{15}$N uptakes distribution ratio in each part of wheat plant.

The Ndffs of the same wheat parts in different saline soil treatments were different (Table 3). The wheat root Ndff of HS treatment was significantly higher than that of LS treatment, but the wheat root Ndff of MS treatment was not significantly different from those of LS treatment and HS treatment ($p < 0.05$). The Ndffs of wheat stem and leaf in HS treatment were significantly higher than those in MS and LS treatments, and the Ndffs of wheat stem of MS treatment was significantly higher than that of LS treatment except for the wheat leaf Ndff ($p < 0.05$). There were no significant differences of the Ndffs in wheat sheath and whole plant among LS, MS, and HS treatments and of the wheat grain Ndff between HS and MS, but the wheat grain Ndffs of HS and MS treatments was significantly higher than that of LS treatment ($p < 0.05$).

Compared with those of the corresponding wheat parts in LS treatment, the Ndffs of the wheat root, stem, leaf, and grain in HS treatment were significantly increased by 46.70%, 34.21%, 21.48%, and 22.46%, respectively ($p < 0.05$). Meanwhile, the trends of the Ndfs in various wheat parts of different saline soil treatments were opposite to those of Ndff. Above all, the Ndffs of each wheat parts in different saline soil treatments showed an upward trend with the increase of soil salinization degrees, except for the wheat sheath and the whole plant, indicating that the increase of soil salinization degree improved the more N absorption of wheat roots, stems, leaves, and grains from fertilizer source and not from soil source to depress salt stress.

### 3.3. $^{15}$N Residues and Distribution in Each 20 cm Soil Layer in 0~100 cm Soil Profile

The soil $^{15}$N residues in three different saline treatments decreased significantly with the increase of soil profile depth (Figure 1).

The $^{15}$N residues in the 0~20 cm soil layer were higher than those in 20~100 cm soil profile by 0.20~317.08 times ($p < 0.05$). Compared with the upper layer (0~20 cm, 20~40 cm, 40~60 cm and 60~80 cm), the $^{15}$N residues of the lower layer (20~40 cm, 40~60 cm, 60~80 cm and 80~100 cm) in the same treatment decreased by 16.68~74.16%, 26.43~95.91%, 0.92~74.81%, 13.19~56.73%, respectively ($p < 0.05$). The $^{15}$N residues of the 0~20 cm soil layer were significantly different among LS, MS and HS treatments which decreased significantly with the increase of soil salinization.

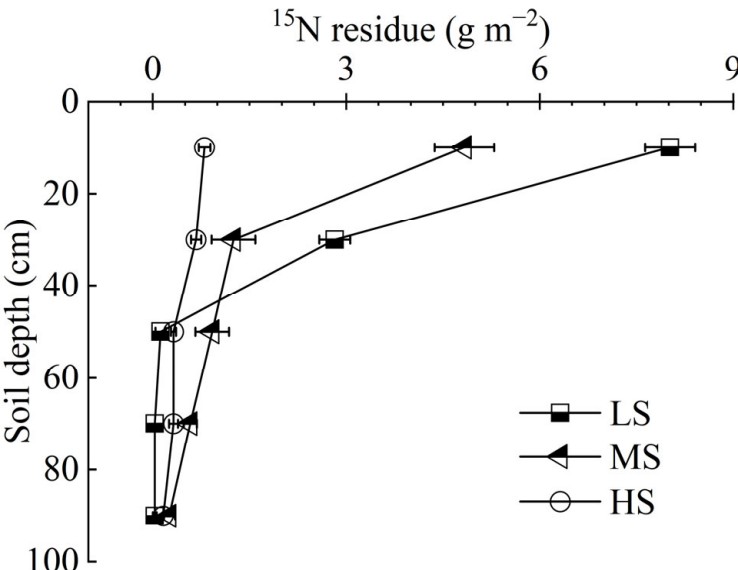

**Figure 1.** $^{15}$N residues in different depth soil layers of different treatments.

The $^{15}$N residues of 0~20 cm soil layer in MS and HS treatments were significantly lower than those in LS treatment by 39.73% and 90.01%, and the $^{15}$N residues of 0~20 cm soil layer in HS treatment were significantly lower than those in MS treatment by 83.42% ($p < 0.05$). The $^{15}$N residues of 20~40 cm soil layer had no significant difference between MS and HS treatments, but both treatments were significantly lower than LS treatment by 55.70% and 76.32% ($p < 0.05$). There was no significant difference in $^{15}$N residues of 40~60 cm soil layers between LS and HS treatments, and both treatments were significantly lower than those of MS treatment respectively by 87.45% and 65.03% ($p < 0.05$). The $^{15}$N residues of 60~80 cm soil layer in MS treatment were higher than those of LS and HS treatments with 1896.16% and 82.07%, and those in HS treatment were higher than those in LS treatment by 996.39% ($p < 0.05$). The $^{15}$N residues of the 80~100 cm soil layer in MS treatment were higher than those in LS treatment by 895.05% except for between both treatments and HS treatment ($p < 0.05$).

Compared with other corresponding treatments at different soil profile depths, the significantly highest $^{15}$N residues in 0~20 cm and 20~40 cm soil layers and in 40~60 cm and 60~80 cm soil layers were LS treatment and MS treatment respectively, but the $^{15}$N residues in 80~100 cm soil layers were not significantly different between HS and LS, and between HS and MS. The above showed that the $^{15}$N residues of different salinization degrees soil treatments mainly remained in 0~20 cm and 20~40 cm, which in other soil layers was significantly reduced. The soil salinization mainly affected the $^{15}$N residues in 0~20 cm and 20~40 cm. With the increase of soil salinization, the $^{15}$N residues in different soil layers also increased significantly with the increase of soil profile depth.

The $^{15}$N residue distribution ratios of 0~20 cm soil layer in three soil treatments with different salinization degrees were significantly higher than those of other soil layers and decreased with the increase of soil profile depth (Figure 2).

With the increase of soil salinization, the $^{15}$N residue distribution ratios in the upper soil layers of 0~100 cm soil profile decreased while those in the lower soil layers increased. However, the $^{15}$N residue distribution ratios in the 0~100 cm soil profile were mainly concentrated in the 0~20 cm soil layer of LS treatment with 72.82%, in the 0~40 cm soil layer of MS treatment with 77.69%, and in the 0~60 cm soil layer of HS treatment with 78.91%, respectively.

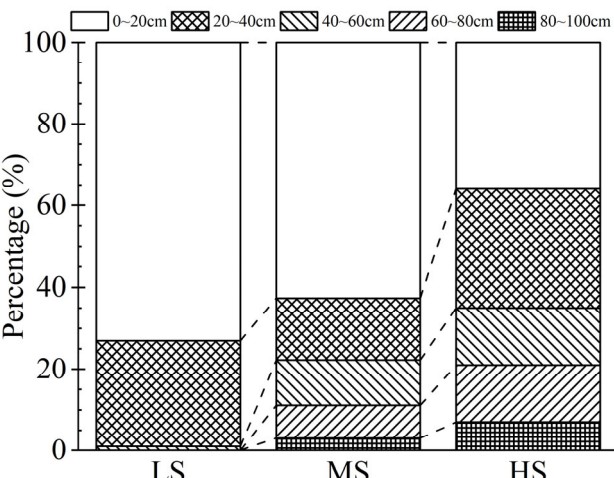

**Figure 2.** $^{15}$N residues distribution ratios in different depth soil layers of different treatments.

The $^{15}$N residue distribution ratio of 0~20 cm soil layer in LS treatment was significantly higher than those in MS and HS treatments respectively by 16.97% and 104.74%, and which in MS treatment was significantly higher than that of HS treatment by 75.04% ($p < 0.05$). The $^{15}$N residue distribution ratio of 20~40 cm soil layer in LS treatment was not significantly different from that in HS treatment, but those of both treatments were significantly higher than that in MS treatment respectively by 66.16% and 89.61% ($p < 0.05$). The $^{15}$N residue distribution ratios of the 40~60 cm soil layer in LS treatment were significantly lower than those in MS and HS treatment respectively by 90.63% and 92.46%, except for that between MS and HS treatment ($p < 0.05$). The $^{15}$N residue distribution ratios of 60~80 cm and 80~100 cm soil layers in LS treatment were significantly lower than those in MS and HS treatments respectively by 96.62% and 98.10%, 93.42% and 97.08%, and those in MS treatment were significantly lower than those in HS treatment respectively by 43.75%, 55.62% ($p < 0.05$). With the aggravation of soil salinization, the $^{15}$N residue distribution ratios of different soil layers in 0~100 cm soil profile increased significantly with the increase of soil profile depth.

The above showed that fertilizer N in saline soil mainly remained in the upper soil layers, such as the topsoil and subsurface layers. However, with the aggravation of soil salinization, the ratio of fertilizer N moving to the deeper soil layer in the 0~100 cm soil profile increased significantly, while the upper soil layer decreased correspondingly.

### 3.4. $^{15}$N Fate in Coastal Saline Soil-Wheat System

After wheat harvest in the field micro-zone, the fates of wheat uptake, soil residue, and system loss of fertilizer nitrogen ($^{15}$N) applied to different saline soil-wheat systems were significantly different (Table 4).

**Table 4.** Balance of $^{15}$N fate in soil-wheat systems of different saline soils treatments.

| Treatment | $^{15}$N Input (g m$^{-2}$) | Plant $^{15}$N Uptake | | Soil $^{15}$N Residue | | System $^{15}$N Loss | |
| --- | --- | --- | --- | --- | --- | --- | --- |
| | | Uptake (g m$^{-2}$) | Uptake Rate (%) | Residue (g m$^{-2}$) | Residual Rate (%) | Loss (g m$^{-2}$) | Loss Rate (%) |
| LS | 22.60 | 3.15 ± 0.28 a | 13.96 ± 1.22 a | 11.00 ± 0.42 a | 48.69 ± 47.15 a | 8.44 ± 0.70 c | 37.35 ± 3.08 c |
| MS | 22.60 | 1.39 ± 0.12 b | 6.17 ± 0.51 b | 7.83 ± 0.98 b | 34.65 ± 4.33 b | 13.37 ± 1.04 b | 59.19 ± 4.60 b |
| HS | 22.60 | 0.35 ± 0.01 c | 1.53 ± 0.04 c | 2.27 ± 0.25 c | 10.05 ± 1.11 c | 19.98 ± 0.25 a | 88.42 ± 1.09 a |

Note: The values in the table are the mean ± standard error and different letters in each column indicate significant difference ($p < 0.05$).

The soil $^{15}$N residues and residual rates were significantly higher than the wheat $^{15}$N uptakes and uptake rates by 248.89%, the system $^{15}$N losses and loss rates by 30.38% in LS

treatment ($p < 0.05$). The system $^{15}$N losses and loss rates in MS and HS treatments were significantly higher than the soil $^{15}$N residues and residual rates of them respectively by 70.84% and 779.78%, and these in MS and HS treatments were significantly higher than wheat $^{15}$N uptakes and uptake rates respectively by 8.60 and 56.73 times, 4.62 and 5.56 times ($p < 0.05$). Therefore, the wheat $^{15}$N uptakes and uptake rates in the different saline soil-wheat systems were all significantly lower than the soil $^{15}$N residues and residual rates and the system$^{15}$N losses and loss rates. The soil $^{15}$N residues and residual rates in lightly salinity soil were significantly higher than the system $^{15}$N losses and loss rates. However, the trend was contrary in medium-salinity and heavy-salinity soils. The aggravation of soil salinization promoted the $^{15}$N loss fate from the saline soil-wheat system.

The wheat $^{15}$N uptakes or uptake rates, the soil $^{15}$N residues or residual rates in HS treatment were significantly lower than those in LS and MS treatment respectively by 89.03% and 75.16%, 79.36% and 70.99%; and those in MS treatment were significantly lower than those in LS treatment respectively by 55.82%, 28.85% ($p < 0.05$). The system $^{15}$N losses or loss rates in HS treatment were significantly higher than those in LS and MS treatment by 136.74% and 49.39%, and this in MS treatments were significantly higher than those in LS treatment by 58.48% ($p < 0.05$). The above showed that with the increase of soil salinization, the system $^{15}$N losses and loss rates increased significantly, while the wheat $^{15}$N uptakes and uptake rates, the soil $^{15}$N residues, and residual rates decreased significantly. With higher degree of soil salinization, the fates of wheat $^{15}$N uptake and soil $^{15}$N residue were reduced, and the fate of $^{15}$N loss was significantly promoted.

## 4. Discussion

The uptake and utilization of nitrogen was an important factor affecting wheats dry matter mass forming through photosynthesis [35]. Our results showed that the order of total N concentration in each part of wheat plant at maturity stage under different salinization degrees soil treatments from high to low was grains, sheaths, leaves, roots, and stems, which was consistent with the result in non-saline soil studied by Wang et al. [36–38]. Our research further found that the total N concentration of wheat roots was between that of leaves and stems, and the order of dry matter masses of wheat plants from high to low was basically as the same as that of total N concentration of wheat plants. Akhtar et al. and Bassil et al. found that the uptake of potassium by crops promoted the N uptake by crops through the joint application of nitrogen and potassium fertilizers [39,40]. Because saline soil has a greater ability to supply potassium, which is rich in available potassium, the total N concentration of each wheat part and the whole plant in our research increased significantly with the aggravation of soil salinization degree.

Our results showed that the dry matter mass accumulation of all wheat parts and the whole plant was inhibited with the fertilizer $^{15}$N uptakes decreased. Soil salinity has adverse effects on crops through two processes: osmotic stress and ion toxicity. Osmotic stress delayed seeds germination, reduced seedling emergence rate and inhibited seedling growth. The high concentration of sodium ions in saline soil rhizosphere resulted in ion imbalance. Sodium and Chloride ions respectively competed with $NH_4^+$ and $NO_3^-$ for crop uptake, inhibited leaves development and photosynthesis, affected the dry matter mass accumulation of crops, and reduced the number and weight of grains [41–45]. When the nitrogen supply of each plant part was limited due to insufficient nitrogen uptake of crops, the dry matter mass of grains might be reduced [46]. Wang et al. found that salt stress inhibited the transport of nitrogen nutrients from vegetative organs (roots and stems) to reproductive organs (grains and sheaths) of crops at maturity, resulting in a decrease of crop grain yield in saline soil [25], which was consistent with the results of our research that the fertilizer $^{15}$N uptakes and its distribution ratios of wheat roots, stems and leaves increased and those of grains decreased in saline soil with aggravation of soil salinization.

Our results showed that the Ndff of wheat grains was 47.1~57.6%, so about half of the N uptake came from fertilizer $^{15}$N. The Ndff of other wheat parts was 25.5~46.1%, and the wheat $^{15}$N uptake mainly came from soil nitrogen. Davis et al. found that the ratio

of fertilizer nitrogen absorbed by crops in the current season was only about 20% in the field crop nitrogen balance experiment [47], and the results of crops preferring to absorb soil nitrogen rather than fertilizer nitrogen were consistent with that of wheat parts except grains in our study. Our study also found that with the aggravation of soil salinization, the ratio of total nitrogen from fertilizer nitrogen in each wheat part increased, while the ratio from soil nitrogen in each wheat part decreased.

Previous studies on the fertilizer nitrogen residue distribution in soil profile showed that the residual rate in different soil layers was inversely proportional to the depth of the soil layer, and the residue of fertilizer nitrogen was mainly concentrated in 0~20 cm soil layer [22], which was consistent with our results. Our study also found that the aggravation of salinization significantly reduced the residues of fertilizer nitrogen in 0~100 cm soil profile and promoted the migration of fertilizer nitrogen to deeper soil layer, which might be one of the reasons for the reduction of fertilizer nitrogen residue in the heavy-salinity soil. Miller et al. had found that the nitrogen accumulation in the soil layer below 20 cm was related to the soil texture [48]. The higher the soil salinity was, the heavier the soil texture was, and the more soil cracks easily resulted in fertilizer nitrogen leaching into the deep soil layer with water [26], which illustrated the results of our study that the fertilizer nitrogen increased significantly with the increase of soil profile depth with the aggravation of soil salinization.

Fertilizer nitrogen applied to saline soil has three main destinations: plant uptake, soil residue, and nitrogen loss in various ways, such as ammonia volatilization, nitrification denitrification, surface runoff, and leaching. Through the study on the fates of fertilizer nitrogen in the saline soil-wheat system, our results found that the current season wheat plants uptake rate of fertilizer nitrogen was only 1.53~13.96% of saline soil in the Yellow River Delta, which was far lower than the average utilization rate of wheat nitrogen in China of 28.2% and the global utilization rate of wheat nitrogen of 30~50% [49]. Our research also found that the utilization of fertilizer nitrogen in the current wheat season was far lower than soil residue and system loss of fertilizer nitrogen, and the aggravation of soil salinization reduced the uptake and residue of fertilizer nitrogen and increase its loss. The above results were because denitrification of nitrogen, volatilization of ammonia, and intensified leaching and migration were the main reasons for less residue and higher loss of fertilizer nitrogen in saline soil [50], and $N_2O$ emission and ammonia volatilization were significantly increased after nitrogen application in saline soils with the aggravation of soil salinity [51].

Therefore, the characteristic of the conventional nitrogen fertilizer application method in the coastal saline soil of the Yellow River Delta was higher nitrogen loss and lower crop uptake and soil residue, which resulted in insufficient nitrogen nutrition for regional crop growth and high yield. Hence, it is recommended to scientifically improve the nitrogen fertilizer application method and strengthen nitrogen management in the coastal saline soil area, so as to reduce fertilizer nitrogen loss, increase crop uptake and soil residue, improve soil nitrogen fertility and crop yield, and achieve high efficiency, high yield, and sustainable development in the coastal saline soil region. Whereas the grades of different soil salinization degrees in our study were relatively less, the fates of fertilizer nitrogen of more soil salinization grades should be deeply studied in order to establish the optimal mode of the utilization and management of fertilizer nitrogen in the coastal saline soil-wheat system in the Yellow River Delta.

## 5. Conclusions

Under the coastal saline soil conditions in the Yellow River Delta, the aggravation of soil salinity from 0.2% to 1% significantly inhibited the dry matter masses accumulation and the [15]N uptakes of various wheat parts and whole plant but promoted the total N concentrations of various wheat parts, especially in wheat roots, stems, and leaves parts absorbed from soil. The dry matter masses and total N concentrations of wheat grains in saline soil were significantly higher than those in other wheat parts. The aggravation of soil

salinity significantly increased the $^{15}$N distribution ratios of wheat roots, stems, and leaves, but decreased that of wheat grains, and significantly increased the Ndffs of wheat roots, stems, leaves, and grains, but had no significant effects on the order from high to low of $^{15}$N distribution ratios in various wheat parts. The fates of $^{15}$N applied in the coastal saline soil-wheat system were wheat uptake 1.53~13.96%, soil residue 10.05~48.69%, and $^{15}$N loss 37.35~88.42%. Compared the lighter salinity soil, the heavier salinity soil reduced the $^{15}$N fate of wheat uptakes and soil residues by 89.03% and 79.36%, but significantly increased the $^{15}$N fate of losses by 136.74% in the coastal saline soil-wheat system. Therefore, the conventional nitrogen management of the coastal saline soil in the Yellow River Delta should be optimized to reduce the nitrogen loss and to improve the productivity of the soil-wheat system to ensure food security.

**Author Contributions:** Methodology, X.G. and P.H.; Investigation, Q.Y.; Data curation, F.D.; Writing—original draft, K.Z.; Writing—review & editing, F.S.; Supervision, Y.Z. and W.C.; Project administration, L.W. and Q.L. All authors have read and agreed to the published version of the manuscript.

**Funding:** This research was supported by National Key Research and Development Program Subproject of China (2021YFD190090205) and Key Research and Development Plan of Shandong Province (Major Scientific and Technological Innovation Project, 2021CXGC010704, 2017CXGC0301).

**Data Availability Statement:** Not applicable.

**Acknowledgments:** We thank the reviewers for their useful comments and suggestions.

**Conflicts of Interest:** The authors declare that they have no known competing financial interests or personal relationships that could have appeared to influence the work reported in this article.

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
