# Peer review of "Fertilizer 15N Fates of the Coastal Saline Soil-Wheat Systems with Different Salinization Degrees in the Yellow River Delta"

_water, doi:10.3390/w14223748_

Round 1
Reviewer 1 Report
This manuscript studies the fertilizer 15N fates of the coastal saline soil-wheat systems with different salinization degrees in Yellow River Delta. The methodology used in the present study is sound and the findings interesting. In my opinion, this manuscript is well-written and provides scientific basis for nitrogen management in coastal saline soil-wheat system. I just have a few points that need to be clarified:
* Line 58-62: the logic of this sentence should be further clarified to facilitate understanding.
*Line 89-90: Although the area is mostly salt wasteland, 100% of the area is farmland, why? Please check this sentence.
* Line 97-101 and Table 1: Where are the different letters in each column which indicate significant differences? Please check and add them.
* Line 112-120: The fertilization is not informative enough and more details of the application of 15N labeled urea are suggested to be provided.
* Line 130-155: Please check the tense in this part of the section for the consistency.
* Line 333: Should the title of 3.4.15 be 15N fates in coastal saline soil-wheat system?
* Line 359-360: Were the fates of wheat 15N uptake and soil 15N residue reduced with lower degree of soil salinization? Please check this sentence.
* Line 369-371: This sentence references to two articles but only annotates one. Please check and revise.
* Line 452-454: The comparison object of this sentence is not clear, so it is suggested to be amended.
The presentation of data in the manuscript is appropriate, the statistical methods used are accurate and the discussion is logical. The language in general is fluent.
Author Response
Thank you very much for your valuable comments on the manuscript. We have deeply checked and revised them with point-by-point responses. Please see the attachment.

Reviewer 2 Report
The present manuscript analyzes the fate and the wheat's absorption rate of N under different salinity scenarios in the plains of the Yellow River Delta, China. The authors found that increases of soil salinity decrease wheat dry mass and promoted wheat soil N uptake l but not the fertilized N. Also, they found that salinization promotes N leaching to deeper soil layers. Finally, the paper suggests to carefully review the amount of N that is use in agriculture in order to improve its efficiency. The manuscript is well written and organized, the message is clear and the topic is important for agriculture management. I have some minor comments that could help to improved the manuscript:
On overall I found the results section too long and sometimes repetitive. Perhaps few solid and condense paragraphs will help the reader. One aspect I found important to clarify is what kind of salts are involved in the salinity (are the chlorides, sulfates?) I assumed they it was mainly sodium chloride but it will be good to clarify it in the text. Also, it would be useful to express the salinity levels also in Ds/m.
Author Response

(The authors gave the same response as above.)
